# Towards Explainable Bilingual Multimodal Misinformation Detection and Localization

## Abstract

The increasing realism of multimodal content has made misinformation more subtle and harder to detect, especially in news media where images are frequently paired with bilingual (e.g., Chinese-English) subtitles. Such content often includes localized image edits and cross-lingual inconsistencies that jointly distort meaning while remaining superficially plausible. We introduce **BiMi**, a bilingual multimodal framework that jointly performs region-level localization, cross-modal and cross-lingual consistency detection, and natural language explanation for misinformation analysis. To support generalization, BiMi integrates an online retrieval module that supplements model reasoning with up-to-date external context. We further release **BiMiBench**, a large-scale and comprehensive benchmark constructed by systematically editing real news images and subtitles, comprising **104,000** samples with realistic manipulations across visual and linguistic modalities. To enhance interpretability, we apply Group Relative Policy Optimization (GRPO) to improve explanation quality, marking the first use of GRPO in this domain. Extensive experiments demonstrate that BiMi outperforms strong baselines by up to **+8.9** in classification accuracy, **+15.9** in localization accuracy, and **+2.5** in explanation BERTScore, advancing state-of-the-art performance in realistic, bilingual misinformation detection. Code, models, and datasets will be released.

## 1 Introduction

The rapid progress of large generative models (Achiam et al., 2023; Rombach et al., 2022; Zhao et al., 2025; Bai et al., 2023a; Guo et al., 2025) has substantially lowered the barrier to producing highly realistic multimodal content (OpenAI, 2023), enabling new creative applications but also heightening the risk of multimodal misinformation, where manipulated images and accompanying text jointly mislead audiences (Qi et al., 2024; Liu et al., 2024b). A recent UNESCO report warns that the widespread adoption of generative-AI tools can be exploited to fabricate convincing visual–textual narratives,[1] thereby creating fertile ground for large-scale multimodal misinformation. Lost in Translation (Quelle et al., 2025) further shows that misinformation frequently crosses language boundaries, underscoring the global risk of cross-lingual diffusion. A striking example occurred in early 2020, when a CDC report confirming the first U.S. case of COVID-19 community transmission[2] was mistranslated on Chinese social media to claim that the virus originated in the United States, fueling public misunderstanding and geopolitical tension. These observations reveal how multimodal misinformation can exploit localized image edits and asymmetric subtitle translations to manipulate public perception across language communities. This creates an urgent need for methods capable of detecting fine-grained multimodal and cross-lingual inconsistencies—capabilities that remain largely underexplored. We therefore formulate the problem as bilingual multimodal misinformation detection: jointly localizing manipulated image regions and identifying cross-lingual inconsistencies while generating faithful natural-language explanations.

Despite notable progress in multimodal misinformation detection, existing approaches remain constrained by a predominant focus on coarse-grained image–text alignment and monolingual settings. First, many systems match images and captions only at the global level (Qi et al., 2024; Liu et al., 2024b), leaving them unable to localize fine-grained, region-level manipulations such as localized

---

[1] https://www.unesco.org/en/articles/new-unesco-report-warns-generative-ai-threatens-holocaust-memory
[2] https://edition.cnn.com/2020/02/27/health/us-cases-coronavirus-community-transmission/index.html

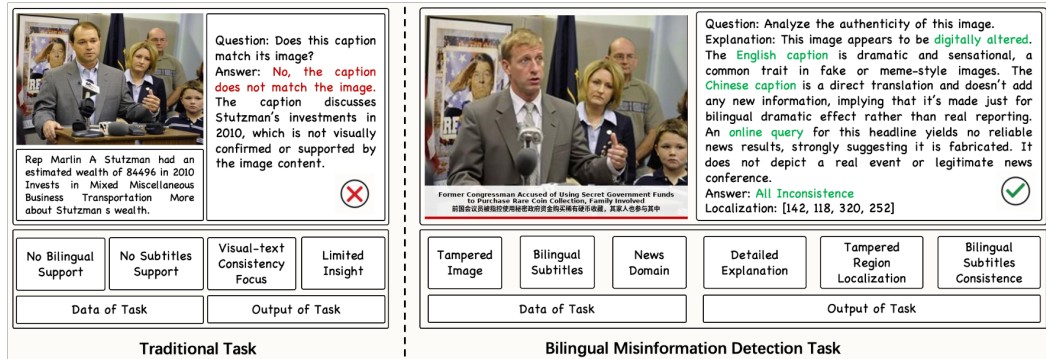

Figure 1: Comparison of traditional vs. bilingual misinformation detection tasks. Traditional tasks focus on visual-text consistency with limited outputs (left). Our setting uses tampered images and bilingual subtitles, enabling richer outputs including region localization, cross-modal consistency, and explanation (right). *Red indicates error, green indicates correctness. Best viewed in color.*

edits or subtle subtitle tampering. Second, their explanation modules typically produce high-level, generic rationales (Shao et al., 2025), providing little concrete evidence of why specific content is misleading. Third, no prior work has specifically addressed bilingual subtitle inconsistencies, leaving the semantic divergence introduced by deliberately misleading cross-lingual translations largely unexplored. These limitations collectively underscore the need for a unified framework that can jointly reason over visual content and bilingual text with precise localization and faithful explanation—capabilities that motivate the approach we develop in this work.

However, building such a framework is technically demanding. Achieving fine-grained grounding between image regions and textual cues remains beyond the capability of most existing multimodal large language models (MLLMs). Moreover, producing faithful explanations in a bilingual multimodal setting requires accurate detection and articulation of inconsistencies across both modalities and languages. Cross-lingual reasoning is further complicated by subtle semantic shifts that even fluent translations can introduce, often obscuring misinformation signals. These challenges are especially acute in the news domain, where content evolves rapidly and exhibits high diversity and context dependence, making it non-trivial to design a system that jointly delivers precise localization, robust cross-lingual reasoning, and faithful explanation.

To address these challenges, we introduce BiMi, the first framework that jointly targets **bilingual subtitles inconsistency**, **region-level manipulation localization**, and **natural-language explanation generation** (Fig. 1). To improve generalization to emerging events, BiMi incorporates an online retrieval module that augments model reasoning with real-time external knowledge. We further construct BiMiBench, the first large-scale benchmark for this setting, containing 104,000 news-image samples with realistic manipulations of visual content and bilingual subtitles. Extensive experiments demonstrate that BiMi establishes a new state of the art, outperforming strong baselines by +8.9 in classification accuracy, +15.9 in localization accuracy, and +2.5 in explanation BERTScore.

Our main contributions are as follows:

- BiMiBench: the first large-scale benchmark for bilingual multimodal misinformation detection, comprising 104K news–image samples with fine-grained visual manipulations and bilingual subtitle inconsistencies.
- BiMi Framework: a unified model that detects image–subtitle misinformation through region-level manipulation localization, multimodal and cross-lingual consistency detection, and natural-language explanation generation, directly addressing key limitations of prior work.
- GRPO for Explanation: the first application of Group Relative Policy Optimization (GRPO) Shao et al. (2024) to improve the quality and faithfulness of bilingual multimodal explanations.
- State-of-the-Art Results: BiMi achieves state-of-the-art performance on BiMiBench, with significant improvements in classification, localization, and explanation BERTScore over strong baselines.

Table 1: Comparison of misinformation datasets. BiMiBench uniquely supports manipulations based on bilingual subtitles and localized content.

| Dataset | Sub-title | Bilin-gual | Textual Manip. | Visual Manip. | Local-ization | News Dom. | Mask |
|---|---|---|---|---|---|---|---|
| FEVER Thorne et al. (2018) | ✗ | ✗ | ✓ | ✗ | ✗ | ✗ | ✗ |
| FakeNewsNet Shu et al. (2020) | ✗ | ✗ | ✗ | ✗ | ✗ | ✓ | ✗ |
| Fakeddit Nakamura et al. (2019) | ✗ | ✗ | ✗ | ✓ | ✓ | ✗ | ✗ |
| MAIM Jaiswal et al. (2017) | ✗ | ✗ | ✓ | ✗ | ✗ | ✗ | ✗ |
| EMU Da et al. (2021) | ✗ | ✗ | ✗ | ✓ | ✓ | ✗ | ✗ |
| NewsCLIPpings Luo et al. (2021) | ✗ | ✗ | ✗ | ✗ | ✗ | ✓ | ✗ |
| COSMOS Aneja et al. (2021) | ✗ | ✗ | ✗ | ✗ | ✗ | ✗ | ✗ |
| DGM4 Shao et al. (2023) | ✗ | ✗ | ✓ | ✓ | ✓ | ✓ | ✗ |
| MMFakeBench Liu et al. (2024b) | ✗ | ✗ | ✓ | ✓ | ✗ | ✗ | ✗ |
| **BiMiBench (Ours)** | ✓ | ✓ | ✓ | ✓ | ✓ | ✓ | ✓ |

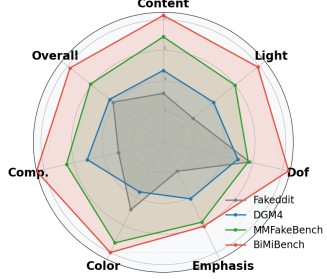

Figure 2: Image quality comparison across datasets using Chen et al. (2024) method.

## 2 RELATED WORK

### 2.1 MISINFORMATION DETECTION

**Datasets.** Early misinformation detection datasets can be grouped into three main categories. Text-only: FEVER (Thorne et al., 2018) and FakeNewsNet (Shu et al., 2020) target textual fact verification and lack visual modality. Basic multimodal: Fakeddit Nakamura et al. (2019), MAIM (Jaiswal et al., 2017), and EMU (Da et al., 2021) pair images with text but are monolingual, use coarse edits, and provide no supervision for localization or explanation. Fine-grained multimodal: NewsCLIP-pings (Luo et al., 2021), COSMOS (Aneja et al., 2021), MMFakeBench (Liu et al., 2024b), and DGM4 (Shao et al., 2023) enable richer reasoning yet still focus on caption-style inputs and ignore subtitle-level or cross-lingual inconsistencies. We introduce BiMiBench, which covers five types of visual and textual misinformation, uniquely supporting bilingual subtitles, region-level localization, and dual-modality tampering (Table 1); a perceptual analysis (Fig. 2) shows it achieves higher realism and diversity than existing datasets.

**Methods.** Current multimodal misinformation methods mainly align image–text pairs for out-of-context detection Przybyla (2020); Qi et al. (2021); Shao et al. (2023). NewsCLIPpings (Luo et al., 2021) and COSMOS (Aneja et al., 2021) use CLIP-based or contrastive learning but lack supervision for fine-grained localization. SNIFFER (Qi et al., 2024) and EMU Da et al. (2021) add explainability via MLLMs yet rely on clean monolingual captions. HAMMER (Shao et al., 2023) localizes manipulations but ignores subtitle-level or multilingual cues. MMD-Agent (Liu et al., 2024b) and CroMe (Choi et al., 2025) broaden evaluation but remain limited to English captions. In contrast, our framework is the first to handle tampered images with bilingual subtitles, supporting localization, cross-lingual reasoning, and explanation.

### 2.2 MULTI-MODAL LARGE LANGUAGE MODELS

**Model.** Multimodal large language models (MLLMs) have greatly advanced cross-modal reasoning. Representative models such as Flamingo (Alayrac et al., 2022), BLIP-2 (Li et al., 2023), MiniGPT-4 (Zhu et al., 2023), Gemini Google (2023), InternVL-3 (Zhu et al., 2025), Qwen-VL (Bai et al., 2023b), and LLaVA (Liu et al., 2023; 2024a) extend pretrained LLMs with cross-modal attention and instruction tuning for vision-language tasks. DeepSeek-R1 (Guo et al., 2025; Shao et al., 2024) further employs GRPO to enhance explanation quality. Yet most MLLMs focus on grounded understanding or generation (Wu et al., 2025; Lin et al., 2025) and are not trained to detect cross-modal inconsistencies. We adapt Gemma 3 (Team, 2025a) to strengthen reasoning over visual and multilingual cues for misinformation detection.

**Training.** MLLMs training typically includes large-scale pre-training followed by post-training with supervised fine-tuning (SFT) and reinforcement learning from human feedback (RLHF)(Ouyang et al., 2022). RLHF aligns outputs with human preferences using a reward model; key methods such as PPO(Schulman et al., 2017), DPO (Rafailov et al., 2023), and GRPO (Guo et al., 2025; Shao et al., 2024) refine policies through preference ranking. RLHF remains little explored for detecting subtle visual edits and bilingual subtitle inconsistencies.

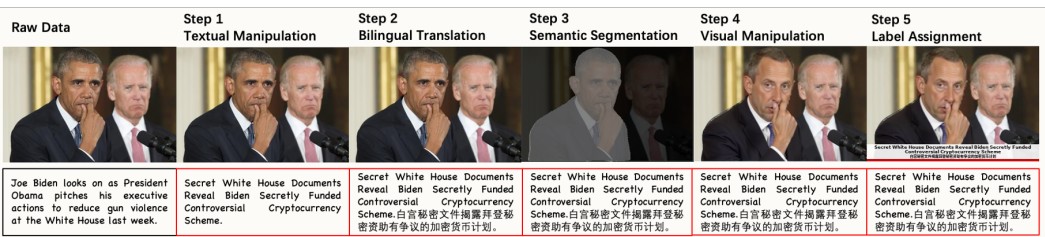

Figure 3: The data generation workflow used in constructing the BiMiBench benchmark.

# 3 BiMiBench: A Benchmark for Bilingual Multimodal Misinformation

Existing benchmarks for multimodal misinformation primarily focus on monolingual captions or synthetic mismatches, lacking realism, cross-lingual scope, and fine-grained supervision. In real-world scenarios, misinformation often involves tampered images with bilingual (Chinese-English) subtitles, where inconsistencies may occur in any modality. We introduce BiMiBench, a benchmark for multimodal misinformation detection with bilingual subtitles, providing labels and explanations for joint evaluation of classification, localization, and explanation quality.

## 3.1 Dataset Construction

**Details.** BiMiBench comprises 104,000 real-world news samples derived from the VisualNews (Liu et al., 2021) corpus, a professionally curated collection of image–text pairs with broad topical diversity and reliable editorial quality. Each sample pairs an image with Chinese–English subtitles; misinformation is introduced via localized image edits or Chinese/English subtitle modifications. About 80% of the samples contain manipulations in at least one modality and 20% serve as clean controls. Images (640×480–1024×768) and bilingual subtitles provide a challenging testbed for evaluating multimodal and bilingual reasoning.

**Construction.** Our benchmark is constructed based on VisualNews, a large-scale image-text news dataset. To generate realistic misinformation samples, we design a multi-step data generation pipeline. Let $\mathcal{D}_{\text{orig}} = \{(I_i, C_i)\}_{i=1}^{N}$ denote the original dataset from VisualNews, where $I_i$ is an image and $C_i$ is its associated English caption.

*Step 1: Textual Manipulation.* Given an image-caption pair $(I_i, C_i)$ from VisualNews Liu et al. (2021), we use the Gemma 3 Team (2025a) model $f_{\text{Gemma}}$ to generate a manipulated caption $\tilde{C}_i = f_{\text{Gemma}}(I_i, C_i, \text{PROMPT})$, introducing inconsistencies while preserving contextual plausibility.

*Step 2. Bilingual Translation.* We translate the original and manipulated English captions into Chinese using a translation API Google Cloud (2024) $\mathcal{T}$, selected for its accuracy and reliability in large-scale bilingual news translation, yielding bilingual subtitle pairs: $C_i^{\text{zh}} = \mathcal{T}(C_i), \tilde{C}_i^{\text{zh}} = \mathcal{T}(\tilde{C}_i)$.

*Step 3. Semantic Segmentation.* To enable targeted visual editing, we use an instruction-based segmentation model LISA Lai et al. (2023):$f_{\text{LISA}}$ to extract object masks based on the manipulated caption: $M_i = f_{\text{LISA}}(I_i, \tilde{C}_i, \text{PROMPT})$, where $M_i$ gets regions corresponding to entities in $\tilde{C}_i$.

*Step 4. Visual Manipulation.* We apply a visual manipulation function $\mathcal{V}$ to perform localized editing on the original image using the object masks: $\tilde{I}_i = \mathcal{V}(I_i, M_i)$, where the edited image $\tilde{I}_i$ semantically aligns with the manipulated caption $\tilde{C}_i$. To ensure visual diversity and realism, we adopt a mix of recent state-of-the-art image editing techniques, including FLUX Labs (2024), VAR Tian et al. (2024), and SDXL Podell et al. (2023).

*Step 5. Label Assignment.* For each final sample, we randomly select real or manipulated versions of the image and subtitles to construct a multimodal example: $S_i^* = S\left(I_i^*, C_i^{\text{en}}, L_i^{\text{zh}}\right), I_i^* \in \{I_i, \tilde{I}_i\}, C_i^{\text{en}} \in \{C_i, \tilde{C}_i\}, L_i^{\text{zh}} \in \{C_i^{\text{zh}}, \tilde{C}_i^{\text{zh}}\}$, where misinformation may appear in any modality individually or in combination.

Figure 4: The overview of the training strategy. Three stages: domain alignment on news data, instruction tuning with task-specific prompts, and GRPO optimization with structured rewards.

**Categories.** Each BiMiBench sample is labeled into one of six categories according to consistency among image $I_i^*$, English subtitle $C_i^{\text{en}}$, and Chinese subtitle $L_i^{\text{zh}}$. (1) *All Consistent*: all modalities agree; (2) *Image Manipulated*: image tampered, at least one subtitle mismatched; (3) *Both Misaligned*: image real, both subtitles misleading; (4) *Chinese Misaligned*: only Chinese subtitle misleading; (5) *English Misaligned*: only English subtitle misleading; (6) *All Inconsistent*: image tampered and both subtitles misleading. Formal definitions show in Table 2.

Table 2: BiMiBench Category Definitions.

| Category | Definition |
|---|---|
| All Consistent | $I_i^* = I_i,\ C_i^{\text{en}} = C_i,\ L_i^{\text{zh}} = C_i^{\text{zh}}$ |
| Image Manipulated | $I_i^* = \tilde{I}_i,\ C_i^{\text{en}} \neq \tilde{C}_i$ or $L_i^{\text{zh}} \neq \tilde{C}_i^{\text{zh}}$ |
| Both Misaligned | $I_i^* = I_i,\ C_i^{\text{en}} = \tilde{C}_i,\ L_i^{\text{zh}} = \tilde{C}_i^{\text{zh}}$ |
| Chinese Misaligned | $I_i^* = I_i,\ C_i^{\text{en}} = C_i,\ L_i^{\text{zh}} = \tilde{C}_i^{\text{zh}}$ |
| English Misaligned | $I_i^* = I_i,\ C_i^{\text{en}} = \tilde{C}_i,\ L_i^{\text{zh}} = C_i^{\text{zh}}$ |
| All Inconsistent | $I_i^* = \tilde{I}_i,\ C_i^{\text{en}} = \tilde{C}_i,\ L_i^{\text{zh}} = \tilde{C}_i^{\text{zh}}$ |

All samples were manually reviewed to ensure annotation quality and consistency. Each item was independently checked by two annotators with a senior reviewer resolving disagreements, following predefined guidelines on factual correctness and translation fidelity. To address potential ethical and privacy concerns, we used only publicly available news content, removed any personally identifiable information, and release the dataset solely for non-commercial research in accordance with the source licenses. This design enables structured supervision of complex multimodal misinformation scenarios while maintaining high standards of data integrity and ethical compliance. Detailed information about the BiMiBench can be found in Appendix A.

# 4 BiMi: A Bilingual Multimodal Misinformation Detection Framework

Real-world multimodal misinformation often involves image edits and inconsistencies between bilingual subtitles. Existing models struggle to detect such cross-modal manipulations and lack adaptability to emerging events. We propose BiMi, a bilingual multimodal framework that localizes manipulated regions, detects cross-modal inconsistencies, and generates explanations. To enhance generalization, BiMi integrates an online retrieval module that provides real-time external context.

## 4.1 Framework

**Modeling.** We adopt Gemma 3 Team (2025a) as BiMi's backbone for its strong multilingual understanding and vision–language alignment, enabling detection of subtle manipulations across images and bilingual subtitles. The input image with overlaid Chinese and English subtitles is encoded into patch-level embeddings, which are fused through attention layers to create a unified multimodal context. This design allows joint reasoning over visual and bilingual textual cues to identify object manipulation, semantic shifts, and cross-lingual inconsistencies.

**Retrieval Module.** To enhance adaptability to emerging misinformation, BiMi employs a retrieval module at inference. Chinese and English subtitles are extracted via OCR to form a bilingual query to the Google Search API (Google Cloud, 2024); the top-3 retrieved passages are prepended to the input as a unified prompt $\mathcal{P} = \text{concat}(R, I, S)$, where $R$, $I$, and $S$ denote the retrieved context, image, and subtitles (image pixels). This auxiliary context supplies timely external information that

improves the model's generalization to previously unseen or rapidly evolving misinformation, while the final predictions remain grounded in the image–subtitle content.

## 4.2 MULTI-STAGE TRAINING

To effectively adapt the pretrained Gemma to the task of multilingual multimodal misinformation detection, we adopt a three-stage training strategy aimed at progressively aligning the model with the news domain, task-specific instructions, and high-quality reasoning objectives. Fig. 4 shows the overview of the training strategy.

**Stage 1: Domain Alignment.** To adapt the model to the linguistic and visual traits of the news domain, we perform instruction-based tuning on VisualNews Liu et al. (2021) image–caption pairs. Each instance uses an instruction prompt (e.g., `Describe this image in English/Chinese`) with an English or translated Chinese caption. This trains the model to generate captions from images and prompts, strengthening visual–text grounding and bilingual reasoning for downstream misinformation detection.

**Stage 2: Instruction Tuning.** We fine-tune the model with single-turn instruction-following data tailored for multimodal misinformation detection. Each sample includes an image $X_v$, a bilingual subtitle pair, and a prompt $X_q$ that asks the model to perform multiple sub-tasks: detect manipulation, assess cross-modal consistency, and explain the decision. The assistant response $X_a$ includes structured answers and a natural language explanation. We follow a unified sequence format:

$$X = \texttt{<system>}\,[X_q; X_v]\,\texttt{<STOP>}\,X_a\,\texttt{<STOP>}$$

where $X_q$ contains the natural language instruction and bilingual subtitles, and $X_a$ consists of three binary labels and a free-form explanation, formatted as:

$$\texttt{<think>}\,E\,\texttt{</think>}\quad\texttt{<answer>}\,y\,\texttt{</answer>}$$

The model is trained to generate $X_a$ conditioned on $X_q$ and $X_v$, by maximizing the likelihood:

$$P(X_a \mid X_q, X_v) = \prod_{i=1}^{L} P_\theta(x_i \mid x_{<i}) \tag{1}$$

where $x_i$ denotes the $i$-th token in the assistant response. Only the tokens in $X_a$ are used to compute the loss. This tuning step encourages the model to align with the structure and reasoning required for fine-grained misinformation detection.

**Stage 3: GRPO-based Reasoning Optimization.** To enhance reasoning and explanation quality for multimodal misinformation detection, we adopt Group Relative Policy Optimization (GRPO) Shao et al. (2024). GRPO ranks candidate outputs within each batch, which fits our setting where explanations for subtle visual edits and bilingual subtitle inconsistencies may be partially correct, and avoids the reward-model sensitivity of PPO and the pairwise preference assumption of DPO.

Given a question $q$, GRPO samples $N$ candidate responses $\{r_1, r_2, \ldots, r_N\}$ from the policy $\pi_\theta$ and evaluates each response $o_i$ using a reward function $R(q, o_i)$, which measures the quality of the candidate in the context of the given question. GRPO encourages the model to generate responses with higher advantages within the group by updating the policy $\pi_\theta$ using the following objective:

$$\mathcal{J}_{\mathrm{GRPO}}(\theta) = \mathbb{E}\left[\{o_i\}_{i=1}^{N} \sim \pi_{\theta_{\mathrm{old}}}(q)\right] \frac{1}{N}\sum_{i=1}^{N}\left\{\min\left[s_1 \cdot A_i,\; s_2 \cdot A_i\right] - \beta\mathbb{D}_{\mathrm{KL}}\left(\pi_\theta \| \pi_{\mathrm{ref}}\right)\right\} \tag{2}$$

$$A_i = \frac{r_i - \mathrm{mean}\{r_1, r_2, \ldots, r_N\}}{\mathrm{std}\{r_1, r_2, \ldots, r_N\}}, \quad s_1 = \frac{\pi_\theta(o_i \mid q)}{\pi_{\theta_{\mathrm{old}}}(o_i \mid q)}, \quad s_2 = \mathrm{clip}\left(s_1,\; 1+\epsilon,\; 1-\epsilon\right), \tag{3}$$

where $A_i$ represents the advantage of the candidate response $o_i$ relative to other sampled responses. Following DeepSeek-R1, we use both format and accuracy reward.

**Reward function.** To optimize the reasoning ability and explanation quality of BiMi during the final training stage, we design a composite reward function tailored for GRPO. In our setting, we

focus on three core tasks: misinformation classification, tampered region localization, and natural language explanation generation.

*Format reward.* To enforce structured outputs, we define a format reward $R_{\text{format}}$ that equals 1 when the model output follows the predefined format with `<answer></answer>` and optional `<think></think>` tags, and 0 otherwise: $R_{\text{format}} = \mathbb{1}[\text{output matches expected format}]$. *Localization reward.* We define an IoU-based reward $R_{\text{Loc}} = \frac{|\mathcal{M}_{\text{pred}} \cap \mathcal{M}_{\text{gt}}|}{|\mathcal{M}_{\text{pred}} \cup \mathcal{M}_{\text{gt}}|}$, where $\mathcal{M}_{\text{pred}}$ and $\mathcal{M}_{\text{gt}}$ are the predicted and ground-truth bounding boxes of the tampered region. *Classification reward.* We define $R_{\text{cls}} = \mathbb{1}[C_{\text{pred}} = C_{\text{gt}}]$. to give a reward of 1 when the predicted label matches the ground truth and 0 otherwise.

The final reward combines all task-specific objectives, including formatting, classification, localization: $R_{\text{total}} = R_{\text{format}} + R_{\text{cls}} + R_{\text{loc}}$. This unified reward encourages the model to generate structured, accurate, and interpretable predictions across modalities without introducing any additional weighting coefficients, so each objective contributes equally.

This progressive training strategy equips BiMi with the capability to perform fine-grained classification, localization, and explanation, and is designed to support generalization to challenging, real-world misinformation cases.

## 5 EXPERIMENTS

### 5.1 EXPERIMENTAL SETUP

We evaluate BiMi on BiMiBench and MMFakeBench (Liu et al., 2024b). On BiMiBench, the model performs *six-class* misinformation classification, region-level *tamper localization*, and *explanation generation*. For MMFakeBench, which does not provide subtitles, we follow its standard *four-class* setting and supply minimal textual context via a unified prompt; subtitle-specific objectives are disabled while the rest of the pipeline remains unchanged.

**Implementation Details.** We adopt GEMMA-3 as the base multimodal large language model and apply the proposed three-stage post-training pipeline: domain align, supervised fine-tuning (SFT) and GRPO-based reinforcement learning. The vision encoder is kept frozen throughout post-training, while only the multimodal adapter and language-model parameters are updated. All experiments are conducted on 48 GB NVIDIA V100 GPUs with a global batch size of 16 and an initial learning rate of $1 \times 10^{-5}$. SFT is run for 3 epochs with early stopping based on validation loss, followed by 2 additional epochs of GRPO training. We fix the random seed to 42 and use AdamW with a weight decay of $1 \times 10^{-2}$. The final reward combines the format, classification, and localization components without introducing any weighting coefficients, i.e., $R_{\text{total}} = R_{\text{format}} + R_{\text{cls}} + R_{\text{loc}}$, so that each objective contributes equally.

**Baselines.** We compare BiMi against a range of strong multimodal baselines, including InternVL3 (Zhu et al., 2025), Qwen3 (Team, 2025b), LLaVA-1.6 (Liu et al., 2024a), and LLama3.1 (Grattafiori et al., 2024), which represent leading approaches in vision-language modeling. We also include specialized misinformation detection systems such as SNIFFER (Qi et al., 2024) and MMD-Agent (Liu et al., 2024b), designed for out-of-context and multimodal misinformation. All models are evaluated using the same inputs: the image with overlaid Chinese-English subtitles and a task-specific prompt.

**Evaluation Metrics.** We report *Accuracy* (ACC) and *F1* for classification and *IoU* for tampered-region localization. For explanation quality, we follow Liu et al. (Liu et al., 2023) and compute *BERTScore* (Zhang et al., 2020) between model outputs and pseudo-references generated by GPT-4o (Hurst et al., 2024). Pseudo-references are produced via structured prompting with label-grounded templates and manually verified by five trained annotators to ensure factual correctness and strict alignment with the ground-truth manipulations. To confirm that BERTScore reflects human judgment, the same annotators also rated 400 randomly sampled explanations on a 5-point Likert scale across five ground-truth–aligned dimensions—*subtitle alignment*, *visual detail consistency*, *reasoning consistency*, *clarity and readability*, and *completeness*. The high agreement between these human ratings (Fleiss' $\kappa = 0.71$) demonstrates the reliability and practical relevance of our evaluation protocol. More evaluation details are provided in the Appendix B.

Table 3: BiMiBench results (metrics in %). IoU shown when region-level localization is available("–" not applicable). Three-run average ($\pm std < 0.5\%$). $\dagger$: statistically significant vs. the best baseline ($p < 0.05$). *Bold and underline denote best and second-best results.*

| Method | MLLM | Res. | ACC | F1 | IoU | BERT |
|--------|------|------|-----|-----|-----|------|
| InternVL3 Zhu et al. (2025) | Qwen2.5-7B | 224 | 20.85 | 18.42 | 7.23 | 71.82 |
| InternVL3 Zhu et al. (2025) | Qwen2.5-14B | 224 | 28.47 | 26.71 | 14.38 | 76.54 |
| Qwen3 Team (2025b) | Qwen3-8B | 224 | 21.43 | 14.96 | 6.28 | 73.92 |
| LLaVA-1.6 Liu et al. (2024a) | Vicuna-7B | 224 | 22.91 | 20.74 | 8.91 | 75.77 |
| Llama 3.1 Grattafiori et al. (2024) | Llama-3.1-8B | 224 | 24.61 | 21.94 | 5.27 | 72.91 |
| Gemma3 Team (2025a) | Gemma3-4B-IT | 336 | 17.48 | 11.73 | 4.29 | 70.84 |
| SNIFFER Qi et al. (2024) | Vicuna-13B | 224 | 37.90 | 33.63 | – | 80.41 |
| MMD-Agent Liu et al. (2024b) | Qwen2.5-7B | 224 | 34.78 | 29.48 | – | – |
| **BiMi** | Gemma3-4B-IT | 336 | **46.80**$^\dagger$ | **42.79**$^\dagger$ | **30.24**$^\dagger$ | **82.90**$^\dagger$ |

## 5.2 Main Results

**Performance Comparison.** We compare BiMi with baselines across three subtasks on the BiMiBench test set: (1) misinformation detection, (2) tampered region localization, and (3) explanation generation. Results are summarized in Table 3. BiMi outperforms all baselines across all tasks. In particular, it achieves a +8.9 accuracy gain in image-subtitle consistency, a +15.9 localization accuracy, and a +2.5 BERTScore improvement in explanation generation over the strongest baseline. BiMi's performance also benefits from the GRPO reward design, which improves explanation specificity and localization precision. The results on MMFakeBench (MMFB, 4-class classification task) are shown in Table 4.

Table 4: MMFB results.

| Method | ACC | F1 |
|--------|-----|-----|
| InternVL3 | 42.69 | 34.80 |
| Qwen3 | 44.82 | 39.49 |
| LLaVA-1.6 | 30.09 | 24.31 |
| LLama3.1 | 32.67 | 28.30 |
| Gemma3 | 27.19 | 22.04 |
| SNIFFER | 67.49 | 61.44 |
| MMD-Agent | 62.78 | 52.83 |
| **BiMi** | **70.94** | **62.49** |

**Ablation Study.** We investigate the contributions of each stage in our pipeline using an ablation study. (i) *Domain Alignment.* Removing this stage reduces accuracy by 2.35 and BERTScore by 4.48, indicating its role in grounding visual-textual reasoning on news content. (ii) *Instruction Tuning.* Eliminating instruction tuning leads to a drastic performance drop, accuracy falls by 31.18 and BERTScore by 12.06, highlighting its importance in enabling structured bilingual understanding. (iii) *GRPO.* Without GRPO, the model fails to generate specific explanations tied to image or subtitle content, reducing accuracy by 5.7 and BERTScore by 8.09. (iv) *Output Order.* Reversing the output order (classify before explain) slightly lowers accuracy (–1.94) and BERTScore (–1.11), suggesting that generating explanations first facilitates more coherent reasoning. Table 5 summarizes the results. These findings confirm that all components are essential, with instruction tuning and GRPO being particularly critical for boosting accuracy and explanation quality.

Table 5: Table: Ablation results (%). Red text shows a performance drop from the best variant.

| Design | Accuracy | BERTScore |
|--------|----------|-----------|
| Full model | **48.60** | **82.90** |
| w/o Domain Align | 46.25 (-2.35) | 78.42 (-4.48) |
| w/o Instr. Tune | 17.42 (-31.18) | 70.84 (-12.06) |
| w/o GRPO | 42.90 (-5.70) | 74.81 (-8.09) |
| Answer First | 46.66 (-1.94) | 81.79 (-1.11) |

## 5.3 Explainability Analysis

Beyond accurate classification and localization, generating high-quality natural language explanations is crucial for interpretable misinformation detection. We analyze explanation ability through qualitative comparisons, the influence of the retrieval module, and evaluation under bilingual.

**Explanation Quality.** We compare BiMi with InternVL3 (Zhu et al., 2025) on a representative case (Fig. 5). Both models capture the bilingual subtitle alignment, but InternVL3 relies on external cues and misses the image–text mismatch. BiMi instead grounds its decision in the image, correctly localizes the manipulation, and offers a clearer step-by-step explanation.

Figure 5: Comparison of explanations from InternVL3 (middle) and our model (right). Top left: input; bottom left: original sample. *Some responses are truncated due to space constraints.*

**Retrieval Module on Real Samples.** We further tested the retrieval component on a 100-sample set of authentic bilingual news images (not included in BiMiBench) to examine its behavior under real-world conditions. Human evaluation reveals that retrieved evidence enhances explanation clarity or correctly resolves subtle cross-lingual manipulations in 9% of cases, remains neutral in 89%, and produces off-topic passages in only 2%, typically due to OCR noise in Chinese subtitles. Even in these rare failures, the final predictions are unaffected, showing that the retrieval module provides timely factual context and improves explanation quality while keeping negative impact minimal.

**Bilingual multimodal capability.** To evaluate bilingual understanding, we construct a variant of the test set where the model receives only one subtitle, Chinese or English. This reduces the task to a 4-category classification. By ignoring cross-lingual consistency, the single-language setting reduces reasoning difficulty. As shown in Table 6, the model performs better with English-only inputs, likely due to the predominance of English data during training. This highlights a gap in bilingual support and the need for stronger Chinese-language modeling in this domain.

Table 6: Accuracy and BERTScore for each language variant (4-category classification).

| Input Variant | ACC | BERTScore |
|---|---|---|
| CN-only | 72.32 | – |
| EN-only | 82.48 | 83.44 |

**Failure Cases.** We observe that BiMi occasionally fails when manipulations are extremely subtle or when OCR fails to extract accurate subtitles. In such cases, either the localization is off-target, or the explanation is overly general. More failure cases and analysis are provided in the Appendix C.6.

## 6 CONCLUSION AND LIMITATION

**Conclusion.** We study the task of detecting multimodal inconsistencies in news images with bilingual (Chinese-English) subtitles. To support this, we introduce BiMiBench, a large-scale benchmark with 104K samples featuring realistic visual edits and cross-lingual mismatches. We propose BiMi, a bilingual multimodal framework combining a three-stage post-training strategy and retrieval-augmented reasoning. Experiments show that BiMi achieves state-of-the-art performance in classification, localization, and explanation, highlighting its effectiveness in real-world scenarios. This work provides a contribution toward advancing research on bilingual and multimodal misinformation detection.

**Limitation.** While BiMi achieves strong results on bilingual multimodal detection, several limitations remain. First, it currently targets only Chinese–English subtitles; extending to additional languages would improve cross-lingual generalization. Second, localization is restricted to bounding boxes and may miss fine-grained manipulations. Third, the data diversity of BiMiBench is limited: beyond broader subtitle templates and editing styles, richer sources spanning multiple languages and topical domains are needed to better reflect real-world variability. Fourth, retrieval introduces extra latency and depends on external sources, which can occasionally return noisy or outdated information. Finally, BiMi's reasoning and explanation capability could be strengthened, especially for cases with weak or conflicting multimodal evidence.

ETHICS STATEMENT

All data are from the publicly available VisualNews corpus; all manipulations were synthetically generated for research only and clearly labeled to prevent misuse. No personal or sensitive information is included.

REPRODUCIBILITY STATEMENT

All experimental settings, dataset construction steps and model details are described in the main paper and Appendix. Source code and data splits will be released upon acceptance to allow full reproduction of our results.

USE OF LARGE LANGUAGE MODELS

Large language models (e.g., ChatGPT) were used only for minor language polishing; all technical content and analyses are our own.

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

## APPENDIX

This appendix provides supplementary material supporting the main paper. Appendix A describes the construction of **BiMiBench**, including data sources, manipulation generation strategy, annotation workflow, and ethical considerations. Appendix B provides full **implementation details** of the BiMi framework, covering prompt design, retrieval-module implementation, and GRPO training setup. Appendix C reports **additional experiments**, such as extended ablation studies, cross-dataset evaluations, and qualitative case studies.

## A DATASET CONSTRUCTION AND QUALITY CONTROL

### A.1 DATASET STATISTICS

BiMiBench comprises 104,000 samples spanning six fine-grained misinformation categories. Figure 6 summarizes key dataset properties. The left panel shows the distribution of samples across categories, ensuring diverse coverage of manipulation types.

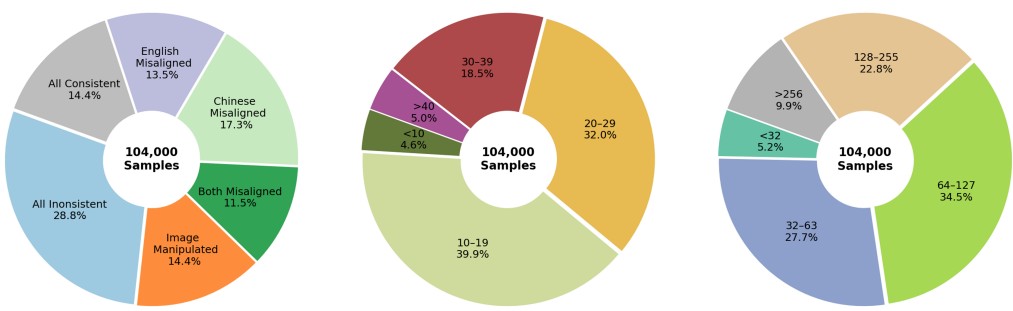

Figure 6: Dataset statistics of BiMiBench. Left: distribution across six misinformation categories. Center: English subtitle length distribution. Right: normalized area of manipulated regions (bounding boxes height).

### A.2 MISINFORMATION GENERATION STRATEGY.

To simulate realistic multimodal misinformation, we generate manipulated subtitles by introducing subtle yet misleading semantic alterations. These edits preserve grammatical fluency and contextual plausibility while modifying key factual elements such as subjects, causal relations, locations, or numerical values. We further introduce bilingual inconsistencies by translating the manipulated English subtitles into Chinese, allowing semantic shifts to emerge naturally during translation. This strategy enables diverse and challenging misinformation scenarios that reflect real-world cross-modal and cross-lingual manipulations. We design task-specific prompts to guide misinformation generation, ensuring fluency and semantic plausibility while introducing factual inconsistencies; an example prompt used in this process is shown in Table 7.

---

**\<system\>**
Based on the image, create a fictional, sensational news headline or summary in one sentence. The story must be clearly fake but sound plausible, and must be no more than 30 words. The news should sound outrageous, scandalous, or unbelievable, yet still written in a serious, news-like tone.
**\</system\>**

---

Table 7: Prompt templates used in BiMi for multimodal misinformation detection

**Data Diversity.** To ensure broad and representative diversity in the synthetic misinformation, we introduce variation at three complementary stages. *Data source*: we begin with the large and professionally curated VisualNews corpus, whose wide topical coverage—politics, economics, sports,

culture—naturally provides a variety of writing styles and visual contexts, ensuring that manipulations are not limited to a narrow set of news domains. *Editing*: on top of this diverse base we generate multiple kinds of factual distortions, ranging from single–word entity swaps to complex changes in temporal references, numerical values and causal relations. We also randomize prompt formulations and the sampling parameters of the generation model so that the manipulated bilingual subtitles exhibit heterogeneous and unpredictable inconsistencies. *Label assignment*: after generation, each candidate sample is manually reviewed by trained annotators and assigned to one of the six defined misinformation categories, reflecting the precise combination of visual edits and cross-lingual subtitle inconsistencies. This three-stage process guarantees that BiMiBench covers a wide spectrum of realistic manipulation patterns and prevents overfitting to any single type of false information.

## A.3 REVIEW PROTOCOL

To ensure data quality and annotation reliability, all samples in BiMiBench were manually verified by a team of five trained annotators, each with backgrounds in journalism, linguistics, or media studies, and prior experience with fact-checking or misinformation analysis. Annotators received targeted instruction on identifying realistic image manipulations, assessing semantic consistency across English-Chinese subtitle pairs, and applying fine-grained manipulation categories. During the review process, they checked the accuracy of tampered regions, validated bounding box alignment, and evaluated bilingual subtitles for cross-lingual fidelity and plausibility. Samples with OCR errors, ambiguous edits, mistranslations, or low visual quality were discarded. This multi-layered, expert-driven verification process ensures that BiMiBench maintains high annotation quality, semantic precision, and supports reliable benchmarking for multilingual multimodal misinformation detection.

## A.4 ETHICAL AND LICENSING CONSIDERATIONS

BiMiBench is derived from the publicly available VisualNews corpus, which provides professionally curated image–text news pairs under a license permitting academic research. All original images and captions remain the intellectual property of their respective news organizations; our release is strictly for non-commercial, research purposes and requires users to acknowledge the original sources and this benchmark in any derivative work.

While BiMiBench is designed to advance scientific understanding of multimodal misinformation detection, it necessarily contains intentionally manipulated content: images edited to introduce subtle visual changes and bilingual subtitles rewritten to create semantic inconsistencies. This "dual nature"—a resource built for research, yet embedding realistic examples of fabricated content—poses an inherent risk of misuse if taken out of context or circulated without clear academic framing.

To mitigate these risks, we (i) clearly mark all manipulated samples as synthetic, (ii) provide the data only for research and educational use, and (iii) require that any redistribution or downstream work include appropriate citations and comply with relevant copyright and data-protection regulations. We explicitly prohibit using BiMiBench to create or disseminate deceptive media outside controlled research settings. These safeguards aim to ensure that the benchmark remains a tool for studying and countering misinformation rather than a source of it.

## A.5 EXAMPLE VISUALIZATION.

Figure 7 illustrates representative BiMiBench samples, covering various manipulation types, language inconsistencies, and corresponding ground truth labels.

## B IMPLEMENTATION DETAILS

### B.1 TRAINING

**Model Architecture.** Our framework is built upon the publicly released **Gemma 3** multimodal language model. The vision encoder is initialized from a pretrained ViT-L and kept frozen in all training stages, while LoRA adapters are fine-tuned on top of the language backbone to reduce computational

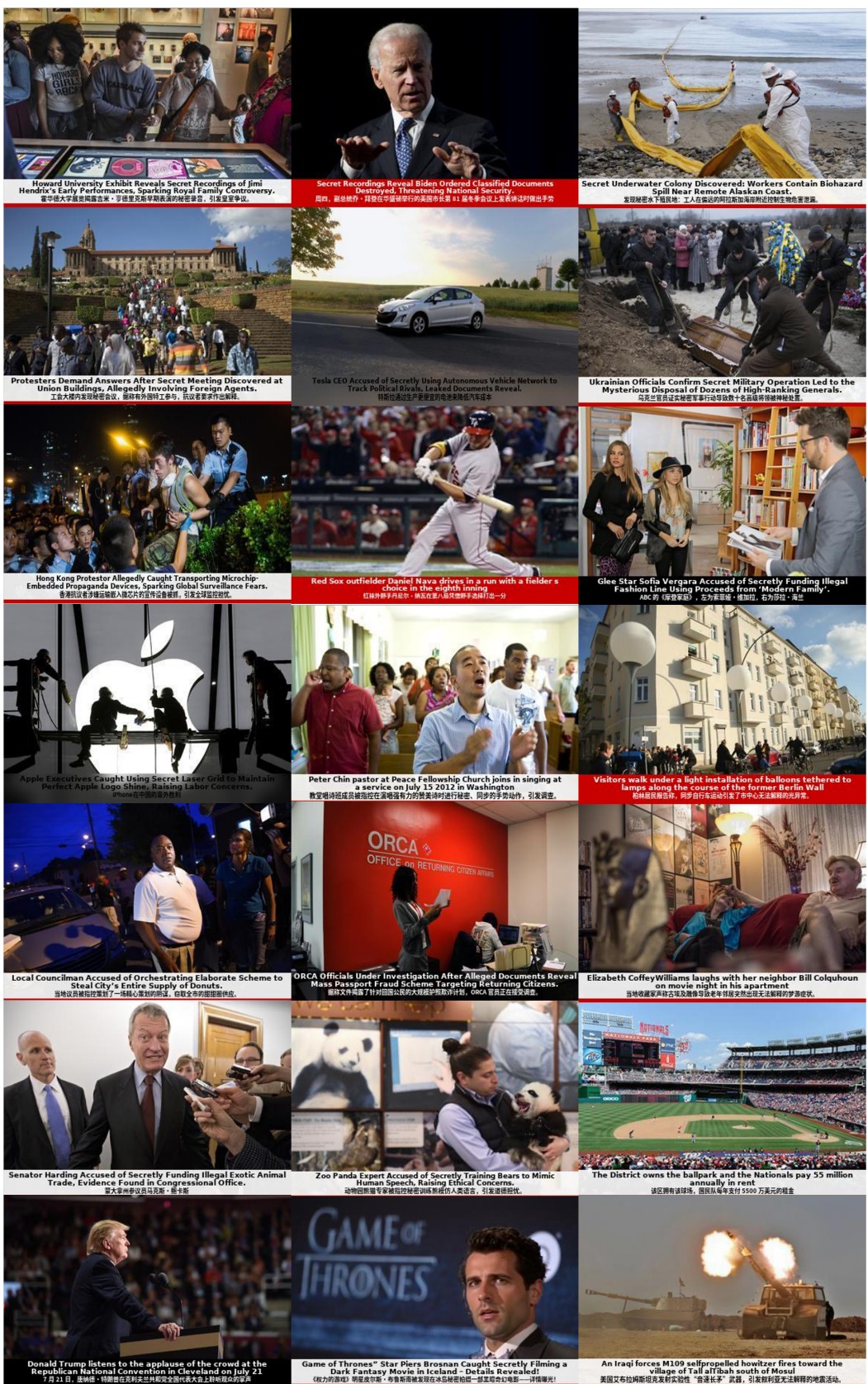

Figure 7: Sample BiMiBench examples with bilingual subtitles and visual or text manipulations.

cost. The model accepts as input the news image with its English–Chinese subtitles, serialized into a unified prompt that embeds the subtitles as text tokens and the image as patch-level embeddings. Visual tokens from the frozen encoder are projected into the language model's embedding space and fused with textual tokens through the cross-modal attention layers of Gemma 3, enabling joint reasoning over visual regions and bilingual text. This design allows efficient adaptation to our task while preserving the strong multilingual and vision–language alignment of the pretrained model.

**Tokenization and Input Format.** Both English and Chinese subtitles are tokenized using the SentencePiece tokenizer aligned with the Gemma 3 backbone to ensure consistent multilingual encoding. The input is formatted into a structured prompt that specifies the prediction task and clearly marks different modalities; visual features from the frozen ViT-L encoder are projected into the same embedding space and appended as a sequence of visual tokens. The combined text and image tokens are then fed to the cross-modal transformer, with the total input length (text + visual tokens) capped at 512 text tokens plus the visual tokens.

**Hyperparameter Summary.** We fine-tune BiMi using the Gemma 3 model (4B) with a batch size of 8, a learning rate of $1 \times 10^{-5}$, and a maximum input length of 512 tokens. The model is optimized with AdamW, using $\beta_1 = 0.9$, $\beta_2 = 0.98$, and a weight decay of 0.01. To stabilize training, we use gradient accumulation (4 steps) and mixed precision (FP16). The GRPO reward coefficient $\lambda$s are all set to 1. These hyperparameters are kept consistent across all three training stages unless otherwise specified.

**Prompt Template.** We design a structured instruction format to guide the model in reasoning across modalities and generating faithful explanations. The prompt template used during instruction tuning and inference is shown in Table 8. This format ensures consistent output structure, supporting multimodal reasoning (via `<think>`), interpretable classification, and region-level manipulation localization.

### B.2 RETRIEVE MODULE

To improve adaptability to rapidly emerging misinformation, we design a retrieval module that operates *only* at inference time and does not require additional training. First, we apply OCR to extract both the Chinese and English subtitles embedded in the news image. The two subtitles are concatenated to form a bilingual query, which is then sent to the Google Search API (Google Cloud, 2024). The API returns up to the top-3 relevant passages; if no relevant documents are found, the retrieval component simply returns an empty string. The retrieved text snippets are concatenated and prepended to the model's input prompt, yielding a unified representation $\mathcal{P} = \text{concat}(R, I, S)$, where $R$, $I$ and $S$ denote the retrieved context, the image and the bilingual subtitles, respectively. This augmented prompt is fed to the MLLM backbone so that cross-attention layers can jointly encode the external evidence and the visual–textual cues. Because the module only provides auxiliary context and does not participate in training, the final predictions remain primarily grounded in the image–subtitle content while benefiting from timely external information when available.

## C ADDITIONAL EXPERIMENTS

### C.1 REFERENCE GENERATION

To construct high-quality reference explanations for automatic evaluation, we adopt a two-stage human–LLM pipeline. First, for each sample we prompt GPT-4o with task-specific, label-grounded templates to generate *two* candidate explanations conditioned on the manipulated modality and misinformation type. Second, a team of five trained annotators—each with a background in journalism, media studies, or NLP—independently reviewed the two candidates and selected the one that best satisfied five ground-truth–aligned dimensions: (i) *Subtitle alignment*, (ii) *Visual detail consistency*, (iii) *Reasoning consistency*, (iv) *Clarity and readability*, and (v) *Completeness*. The chosen explanation for each sample was retained as the pseudo-ground truth used in BERTScore evaluation. This procedure ensures that every reference explanation is both fluent and factually faithful while keeping the generation process efficient and reproducible.

Table 8: Prompt templates used in BiMi for multimodal misinformation detection

<**system**>
You are an expert in misinformation detection area. A conversation between User and Assistant. The user asks a question, and the Assistant solves it. The assistant first thinks about the reasoning process in the mind and then provides the user with the answer. The reasoning process and answer are enclosed within <think> </think> and <answer> </answer> tags, respectively, i.e., <think> reasoning process here </think><answer> answer here </answer>.
Assign one of the following six categories based on their mutual alignment: All Consistent: The image aligns well with both Chinese and English captions. No signs of manipulation or misalignment. Image Manipulated: The image is manipulated. Both captions truthfully describe the manipulated image. Both Misaligned: Both Chinese and English captions are manipulated. Neither caption correctly describes the image. Chinese Misaligned: Only the Chinese caption is manipulated. The English caption aligns correctly with the image. English Misaligned: Only the English caption is manipulated. The Chinese caption aligns correctly with the image. All Inconsistent: The image, Chinese caption, and English caption are all manipulated and mutually inconsistent.
</**system**>
<**user**>
Please analyze the given image containing both Chinese and English subtitles and complete the following three tasks:
(1) Classification Task: classify the alignment between the image and the subtitles into one of the following six categories: `"all consistent"`, `"image manipulated"`, `"both subtitles misaligned with image"`, `"only English aligned"`, `"only Chinese aligned"`, `"all inconsistent"`.
(2) Manipulation Detection: if the image has been manipulated, return one or more bounding boxes for the manipulated regions in the format: `{"bbox":[x_min, y_min, x_max, y_max]}`. If no manipulation is found, return an empty list.
(3) Decision Explanation: briefly explain your thinking before the classification and any detected regions.
Return your output using the following format, wrapped in tags:
<think>
Your explanation here.
</think>
<answer>
`"classification": "result", "region": [{"bbox": [x_min, y_min, x_max, y_max]}]`
</answer>
</**user**>

## C.2 PERFORMANCE ON REAL-WORLD SAMPLES

To assess real-world generalization, we manually collected 100 recent social-media posts spanning political, health, and international news. These posts contain image–caption manipulations, cross-lingual inconsistencies, and subtle visual edits typical of in-the-wild misinformation. Because the authenticity of such posts cannot be verified with ground-truth labels, the 100 examples were *expert-curated*: five annotators with backgrounds in journalism and fact-checking jointly reviewed the content and reached consensus on whether each case constitutes misinformation. This human consensus served as the reference label set.

Evaluated against these expert judgments, BiMi correctly identified 81 of the 100 curated cases (81% accuracy) . These findings suggest that BiMi can transfer effectively from synthetic benchmarks to uncontrolled real-world conditions, even when the ground truth is based on expert assessment rather than definitive factual verification.

We analyze a real-world social media post with subtle bilingual inconsistency (Figure 8). The re-moved "student" in the Chinese translation alters the meaning. Without retrieval, the model mis-

classifies due to a lack of context. The retrieval result provides a key clue for classification. BiMi correctly identifies the inconsistency, showing its benefit for nuanced cross-lingual reasoning.

Figure 8: Social media example with subtle bilingual inconsistency. The Chinese translation omits "student", changing the meaning and potentially misleading Chinese readers. Middle: InternVL3 result; Right: BiMi result.

## C.3 EFFECT OF OCR MODULE

To quantify the impact of subtitle extraction quality, we analyzed BiMi's sensitivity to OCR noise through two complementary studies.

**Failure-case analysis.** A manual review of misclassified samples shows that roughly 37% of errors stem from OCR issues—such as missing or truncated characters in long Chinese subtitles, and occasional rare-character misrecognition in low-resolution or stylized fonts. These failures occur primarily in Chinese inputs, while English subtitles exhibit minimal OCR-related errors.

**Controlled evaluation.** We compared model performance using ground-truth subtitles versus OCR-extracted subtitles.

Table 9: Classification accuracy (ACC) and BERTScore (%) using *Ground-truth* subtitles (manually provided) versus *OCR-extracted* subtitles (automatically recognized).

| Language | Subtitle Source | ACC (%) | BERTScore |
|----------|-----------------|---------|-----------|
| English  | Ground-truth    | 84.2    | 88.1      |
| English  | OCR-extracted   | 83.0    | 85.9      |
| Chinese  | Ground-truth    | 80.0    | 86.5      |
| Chinese  | OCR-extracted   | 75.3    | 80.2      |

English accuracy drops by only 1.2% and BERTScore by 2.2, whereas Chinese drops by 4.7% and 6.3 respectively, indicating that Chinese performance is more sensitive to OCR noise.

**Conclusion.** These findings show that BiMi remains largely robust to moderate OCR noise; the observed degradation originates mainly from preprocessing rather than from limitations in the model's bilingual reasoning. When clean Chinese subtitles are provided, BiMi's accuracy approaches the English benchmark. Improving OCR fidelity—e.g., via confidence-based filtering or multi-OCR consensus—can therefore further enhance overall system reliability in real-world deployments.

## C.4 ANALYSIS OF BILINGUAL PERFORMANCE GAP

BiMi achieves higher accuracy on English-only inputs (82.48% ACC) than on Chinese-only inputs (72.32% ACC). To disentangle the causes of this gap, we conducted a controlled evaluation using ground-truth subtitles versus OCR-extracted subtitles:

The results show that Chinese performance is far more sensitive to OCR noise, with a 4.7% drop compared to only 1.2% for English. Thus, OCR quality in Chinese subtitles is the primary factor behind the overall 10% gap.

Table 10: Classification accuracy (%) using *Ground-truth* subtitles (manually provided) versus *OCR-extracted* subtitles (automatically recognized).

| Language | Ground-truth ACC (%) | OCR-extracted ACC (%) | Drop (%) |
|---|---|---|---|
| English | 84.2 | 83.0 | -1.2 |
| Chinese | 80.0 | 75.3 | -4.7 |

Secondary contributors include (i) training data imbalance: VisualNews contains more English captions, which biases the backbone model toward English, and (ii) language-model priors: although Gemma 3 supports bilingual reasoning, its representations are stronger for English. When clean Chinese subtitles are provided, BiMi's accuracy approaches the English benchmark, confirming that OCR errors are the dominant bottleneck.

To close this gap, future work will focus on (1) enhancing Chinese OCR through confidence-based filtering and multi-OCR consensus, (2) exploring LLM-based subtitle repair to recover noisy outputs, and (3) fine-tuning on more balanced bilingual data. Overall, the performance difference is mainly driven by OCR noise, with data imbalance and model priors as secondary factors.

### C.5 COMPUTATION AND EFFICIENCY

BiMi employs a relatively lightweight multimodal backbone (Gemma 3–4B) to keep training and inference costs manageable. All experiments are conducted on a single NVIDIA V100 48 GB GPU. End-to-end inference latency is determined by three components: model forward pass, OCR subtitle extraction, and external retrieval. When retrieval is disabled, pure model inference averages 320 ms per sample; enabling the full retrieval pipeline—including OCR and Google Search API—introduces an additional 0.8–1.2 s of latency depending on network conditions and hardware. This overhead remains acceptable for offline evaluation and is comparable to other retrieval-augmented MLLMs.

To avoid retrieval becoming a single point of failure, BiMi incorporates a graceful-degradation mechanism: if OCR or web search times out or returns no relevant context, the model skips the retrieved evidence and falls back to pure visual–textual reasoning, keeping classification and explanation performance stable. During training, the vision encoder is frozen and only LoRA adapters are updated, which keeps GPU memory usage below 40 GB and allows a batch size of 16. These design choices ensure that BiMi can be reproduced on standard academic hardware while maintaining reasonable inference latency and memory footprint.

### C.6 FAILURE CASE STUDY

Despite the overall effectiveness of our model, we observed several notable failure cases that highlight its current limitations. One recurring issue involves semantic misinterpretation, particularly when the input contains rare or ambiguous phrases. For example, in a prompt requiring multi-step reasoning or contextual understanding (e.g., "Describe the implication of climate change on coastal microeconomies"), the model occasionally generates generic or overly broad responses that fail to capture the specific nuances of the query. Another type of failure arises in edge cases involving domain-specific knowledge.

Figure 9 shows a failure case where the model misclassified an image manipulation example as "All Inconsistent." The English caption aligns with the image, but the Chinese one introduces a false espionage narrative. Key issues include: misclassification; incorrect caption judgment—only the Chinese caption was wrong; missing bounding box. This highlights weaknesses in manipulation detection, multilingual judgment, and grounding.

### C.7 QUALITATIVE CASES

Figure 10 11 12 13 14 15 16 17 show the examples.

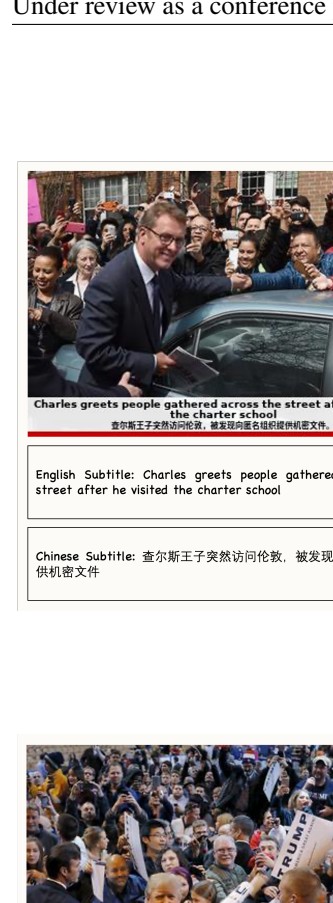

```
<think>
To classify this image and its captions:
Visual Analysis:
    The image shows a man in a suit interacting with a crowd, with people taking pictures and holding
up cell phones. This suggests a high-profile event, possibly involving a public or political figure.
Caption Comparison:
    English Caption: "Charles greets people gathered across the street after he visited the charter
school."
    Chinese Caption: "查尔王子王兹牙尖际间行步访, 致被现向区名组织供密件。"
Caption Interpretation:
    The English caption refers to an event involving "Charles" related to a charter school visit.
    The Chinese caption appears nonsensical and unrelated to the image and doesn't provide coherent
information.
Consistency Check:
    The image portrays a man interacting with the public, which aligns more with political or public
events rather than a school visit. The English caption could align with the visual if "Charles" refers to
a public figure, but it is speculative, and the Chinese caption offers no consistent context. The Chinese
caption contains mistranslation or intentional misrepresentation, making it unrelated to the image.
Conclusion:
    The image doesn't align with either caption because the event depicted seems public-focused, and
the captions are inconsistent in meaning and context. No clear manipulation of the image is visible
(e.g., no altered regions detected). Both captions seem inauthentic or deliberately misleading. Based on
these observations, the classification is "All Inconsistent."
</think>
<answer>
{"classification": "All Inconsistent", "region": {"bbox":[]}}
</answer>
```

Figure 9: Failure case

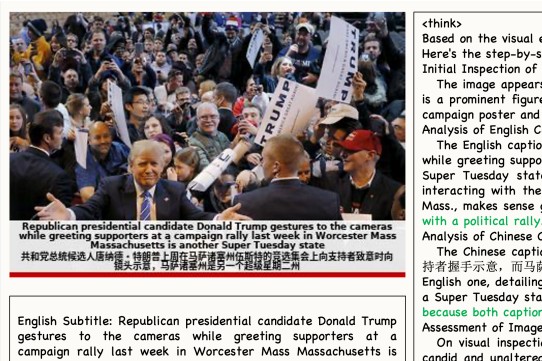

```
<think>
Based on the visual evidence, the alignment between the image and the subtitles needs to be classified.
Here's the step-by-step reasoning:
Initial Inspection of the Image:
    The image appears to depict a campaign rally with a large crowd of supporters. Visible in the image
is a prominent figure in the center, identified as Donald Trump by the presence of a "TRUMP PENCE"
campaign poster and banners.
Analysis of English Caption:
    The English caption states, "Republican presidential candidate Donald Trump gestures to the cameras
while greeting supporters at a campaign rally last week in Worcester, Mass. Massachusetts is another
Super Tuesday state." The scene in the image aligns with this caption as it shows Donald Trump
interacting with the crowd in an environment typical of a campaign rally. The location of Worcester,
Mass., makes sense given the presence of large numbers of people and enthusiasm, which is consistent
with a political rally.
Analysis of Chinese Caption:
    The Chinese caption translates to, "共和党总统候选人特朗普在过去的星期天于马萨诸塞州伍斯特市与支
持者握手示意, 而马萨诸塞州是另一个选举星期二州。" This caption provides the same information as the
English one, detailing that Trump greeted supporters in Worcester, Massachusetts, which is considered
a Super Tuesday state. Consistency between the English and Chinese captions and the image is evident
because both captions describe the same event accurately.
Assessment of Image Integrity:
    On visual inspection, there are no evident signs of image manipulation. The scene depicted appears
candid and unaltered. Details such as the expressions, the number of people, and the items they are
holding align well with what is commonly seen in political rallies.
Conclusion on Alignment:
    Both captions are in alignment with the image content, providing a truthful description of the event.
No discrepancies are observed between the images and the captions. Therefore, the image and the
subtitles are in mutual alignment without signs of manipulation.
</think>
<answer>
{"classification": "All Consistent", "region": {"bbox":[]}}
</answer>
```

Figure 10: Case of All Consistency.

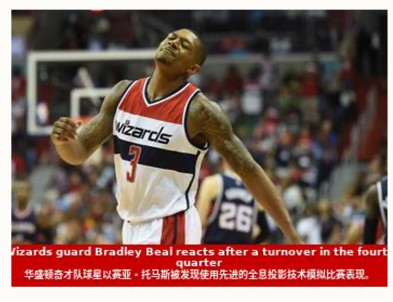

```
<think>
Based on the visual evidence and the provided text analysis, the alignment between the image and the
subtitles needs to be classified. Here's the step-by-step reasoning:
Visual Analysis (Classification):
    The image clearly shows a basketball player in a game. The jerseys and the basketball court in the
background are consistent and intact, with no visible signs of image manipulation. The English subtitle
reads: "Wizards guard Bradley Beal reacts after a turnover in the fourth quarter." This seems to
accurately describe the observed scene, with the player appearing to be in a moment of reaction.
Consistency of Captions (Classification):
    The English caption aligns well with the image content, correctly describing the scene as a basketball
game with a specific player (Bradley Beal) reacting to a turnover. The Chinese caption reads: "华盛顿奇才
队球星以赛亚-托马斯被发现使用先进的全息投影技术模拟比赛表现。" This translates to "Washington
Wizards star Isaiah Thomas is found to be using advanced holographic projection technology to simulate
game performances." This caption does not align with the image since the player in the image is Bradley
Beal, not Isaiah Thomas, and there is no evidence suggesting the use of holographic technology.
Manipulation Detection (Finding Changes in Image):
    The image does not show any signs of manipulation, such as alterations to the player, background, or
text overlays. The bounding box provided in the `region` section should reflect that no manipulation is
detected.
Decision Explanation:
    The primary discrepancy lies in the Chinese subtitle, which inaccurately attributes the actions to a
different player (Isaiah Thomas) and introduces a misleading technology claim that is not supported by
the image. The English subtitle aligns well with the image content, providing accurate information about
Bradley Beal's reaction during a game.
Given the above analysis, the classification of "Chinese Misaligned" is appropriate as the English caption
correctly matches the image, while the Chinese caption does not.
</think>
<answer>
{"classification": "Chinese Misaligned", "region": {"bbox":[]}}
</answer>
```

Figure 11: Case of Chinese Misaligned.

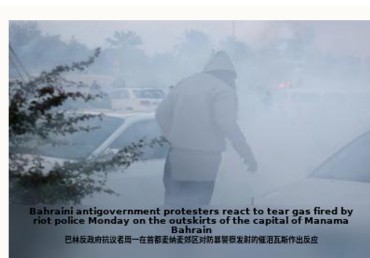

<think>
Visual Analysis of the Image:
    The image depicts a smoky or hazy environment with a person wearing a hooded jacket. The scene appears chaotic, with blurred structures in the background, suggesting a possible tear gas scenario.
Language and Subtitle Analysis:
    The English caption states: "Bahrain antigovernment protesters react to tear gas fired by riot police Monday on the outskirts of the capital of Manama Bahrain." The Chinese caption states: "巴林反对政府抗议者周一在首都市郊对防暴警察发射催泪瓦斯时作出反应。" (Translation: "Bahrain antigovernment protesters react to tear gas fired by riot police Monday on the outskirts of the capital of Manama.")
Consistency Check:
    Both captions mention tear gas and antigovernment protesters reacting to it on the outskirts of Manama on Monday. Considering the hazy environment and the presence of what appears to be a confrontation, the scene aligns with the captions.
Manipulation Detection:
    Analyzing the image for signs of manipulation (e.g., mismatched lighting, unnatural object edges, inconsistencies in the background) is critical. The edges of the person and the surrounding environment display a smooth transition, but closer inspection might reveal anomalies. Given the hazy effect, one potential area of interest is the background where there is significant blur and potential inconsistencies in the smoky elements. This area could be manipulated to simulate the tear gas effect.
Verdict Based on Analysis:
    Both captions are consistent with the visual elements of the image. Based on visual analysis, we identify regions that have been altered to emphasize the tear gas effect, especially in the background area.
Bounding Box for Manipulated Region:
    Focusing on the background where the haze or smoke is most intense and likely manipulated. Bounding box coordinates are determined to encompass the area of potential manipulation.
</think>
<answer>
{"classification": "Image Manipulated", "region": {"bbox":[124,74,208,158]}}
</answer>

English Subtitle: Bahrain antigovernment protesters react to tear gas fired by riot police Monday on the outskirts of the capital of Manama Bahrain

Chinese Subtitle: 巴林反对政府抗议者周一在首都市郊对防暴警察发射催泪瓦斯时作出反应

Figure 12: Case of Image Manipulation.

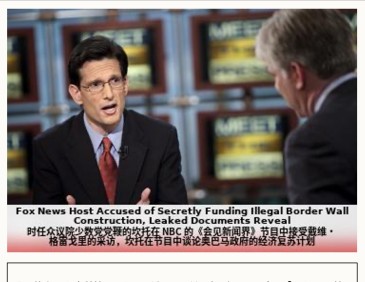

<think>
I need to analyze the given image, which contains both Chinese and English subtitles, and determine the alignment between the image and the captions.
Image Inspection:
    The image shows a man in a suit, likely participating in a news broadcast. Given the context of the English subtitle mentioning "Fox News Host" and "Leaked Documents," the scene seems to fit a news setting.
Subtitle Analysis:
    English Caption: States "Fox News Host Accused of Secretly Funding Illegal Border Wall Construction, Leaked Documents Reveal." Chinese Caption: States similarly about the Fox News host being implicated in secretly funding construction of illegal borders, based on leaked documents.
Subtitles Consistency:
    Although both captions discuss the same scenario involving the Fox News host, they are not semantically consistent. The English caption presents an implausible or exaggerated claim that lacks credibility.
Image Context Verification:
    The image shows a serious discussion but does not specifically indicate the topic of border wall construction or leaked documents. It's possible that the image is not aligned with the specific accusations mentioned in the English caption.
Manipulation Detection:
    The image does not appear to be digitally manipulated. Both captions relate to a plausible news scenario, but the English caption might not match the specific details shown in the image. Combining these observations, it should be "English Misaligned" because the English caption might not accurately reflect the specific details or context shown in the image.
</think>
<answer>
{"classification": "English Misaligned", "region": {"bbox":[]}}
</answer>

English Subtitle: Fox News Host Accused of Secretly Funding Illegal Border Wall Construction, Leaked Documents Reveal

Chinese Subtitle: 时任众议院少数党党鞭的坎托在 NBC 的《会见新闻界》节目中接受戴维·格雷戈里的采访，坎托在节目中谈论奥巴马政府的经济复苏计划

Figure 13: Case of English Misaligned.

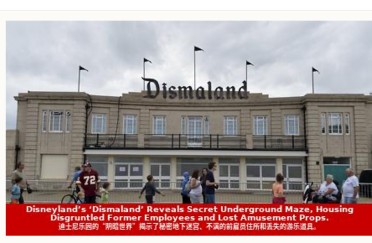

<think>
To determine the category, I analyze both the Image and the accompanying subtitles in English and Chinese. The image shows the entrance of "Dismaland," which is a well-known art installation created by the street artist Banksy in 2015. It was a temporary dystopian theme park in Weston-super-Mare, UK, designed as a satirical take on Disneyland, with clear and deliberate dark humor elements.
English Caption Analysis:
    The English caption claims: "Disneyland's 'Dismaland' Reveals Secret Underground Maze, Housing Disgruntled Former Employees and Lost Amusement Props." This is factually incorrect. Dismaland is not affiliated with Disneyland, and there is no evidence of any secret underground maze, disgruntled employees living beneath it, or lost amusement props being stored underground. This appears to be a fabricated sensational claim.
Chinese Caption Analysis:
    The Chinese caption is a direct translation of the English one: "迪士尼乐园的'阴暗世界'揭示了秘密地下迷宫，不满的前员工和丢失的游乐设施道具。" This too reflects the same fabricated content and does not provide any truthful or accurate description of the image.
Image Analysis:
    The building facade and signage are consistent with the actual appearance of Dismaland as it was constructed during its public exhibition. No apparent image manipulation is observed — the building appears intact and unaltered.
Conclusion:
    The image is not manipulated. Both captions (English and Chinese) provide misleading information and do not accurately describe the image. Hence, the correct classification is Both Misaligned, and no bounding box is needed for manipulated image regions.
</think>
<answer>
{"classification": "Both Misaligned", "region": {"bbox":[]}}
</answer>

English Subtitle: Disneyland's 'Dismaland' Reveals Secret Underground Maze, Housing Disgruntled Former Employees and Lost Amusement Props

Chinese Subtitle: 迪士尼乐园的"阴暗世界"揭示了秘密地下迷宫、不满的前雇员住所和丢失的游乐道具

Figure 14: Case of Both Misaligned.

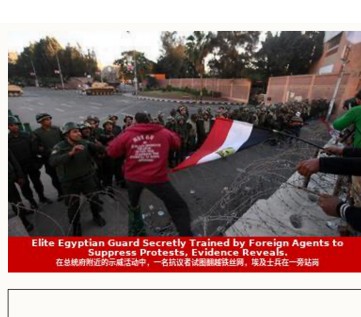

```
<think>
To assess the image and its captions for consistency and potential manipulation, we proceed with a step-by-step analysis:
Image Content:
    The image shows a confrontation scene—presumably from a protest—where a man in a red jacket holding an Egyptian flag appears to be yelling or confronting a line of uniformed soldiers behind a barbed-wire barrier. The environment suggests a high-tension situation, possibly in front of a government building or compound.
English Caption Analysis:
    Claims: "Elite Egyptian Guard Secretly Trained by Foreign Agents to Suppress Protests, Evidence Reveals." This is a bold and specific geopolitical claim that implies covert international training operations, which cannot be visually confirmed or inferred solely from the image content. The image shows a protest and armed guards, but offers no visual indication of "secret training," "foreign agents," or "evidence."
Chinese Caption Analysis:
    Translation: "At a protest near the president's residence, a protester waves the national flag while confronting Egyptian soldiers at the frontline." This is a factual, literal description of what's visible in the image: a protester, Egyptian flag, confrontation with soldiers. There are no unverifiable or speculative geopolitical claims.
Cross-caption Consistency:
    The English caption introduces unverifiable, potentially misleading information about foreign agents and covert operations. The Chinese caption is visually accurate and does not make unverifiable claims.
Image Manipulation Check:
    The image does not appear to be digitally altered; it matches the scene described by the Chinese caption (presence of soldiers, protester, flag, barbed wire, etc.). There is no visual evidence of tampering, overlays, or inserted foreign elements.
Conclusion:
    The image is authentic. The Chinese caption aligns with the image. The English caption is misaligned and introduces external, unverifiable narratives not supported by the image. Therefore, the correct classification is "English Misaligned".
</think>
<answer>
{"classification": "English Misaligned", "region": {"bbox":[]}}
</answer>
```

English Subtitle: Elite Egyptian Guard Secretly Trained by Foreign Agents to Suppress Protests, Evidence Reveals

Chinese Subtitle: 在总统府附近的示威活动中，一名抗议者试图翻越铁丝网，埃及士兵在一旁站岗

Figure 15: Case of English Misaligned.

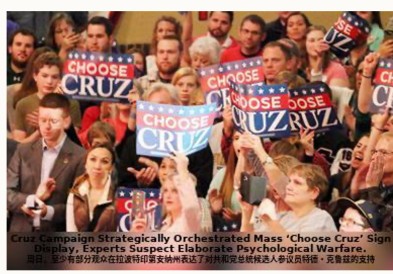

```
<think>
To analyze the alignment between the image and its subtitles, I need to examine whether the image accurately represents the information provided in both the Chinese and English captions.
Visual Analysis:
    The image shows a crowd of people holding signs with "CHOOSE CRUZ."
    The individuals appear to be supporters at a campaign event.
Caption Analysis:
    English Caption: Discusses Cruz's campaign orchestrating a mass "Choose Cruz" sign display and mentions suspicions of psychological warfare.
    Chinese Caption: States that on the 7th, at least some supporters at Cruz's hometown showed their support for Cruz and Cruz's presidential candidate.
Consistency Check:
    The image aligns with the general idea of a campaign event with supporters holding signs, as described in both captions. However, the Chinese caption claims a specific event on the 7th in Cruz's hometown, while the image does not provide specific temporal or locational context that confirms this claim.
Detection of Manipulation:
    The image has been manipulated, but the alteration is limited to the person in the foreground. There are no overt anomalies in the rest of the scene, making the manipulation subtle.
Decision:
    Given the potential discrepancy between the specific claims in the captions and the lack of contextual support in the image, I classify this as "All Inconsistent." This category is chosen because the image, the English caption, and the Chinese caption seem to be mutually inconsistent due to unconfirmed specifics in the captions.
Bounding Box for Manipulated Region:
    Focusing on the person in the foreground likely manipulated. Bounding box coordinates are determined to encompass the area of potential manipulation.
</think>
<answer>{"classification": "All Inconsistent", "region": {"bbox":[116,244,179,289]}}</answer>
```

English Subtitle: Cruz Campaign Strategically Orchestrated Mass 'Choose Cruz' Sign Display, Experts Suspect Elaborate Psychological Warfare

Chinese Subtitle: 周日，至少有部分观众在拉波特印第安纳州表达了对共和党总统候选人参议员特德·克鲁兹的支持

Figure 16: Case of All Inconsistent.

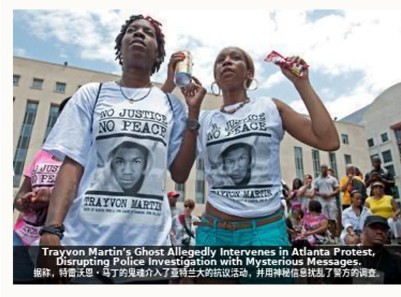

```
<think>
To analyze the alignment between the image and the subtitles, let's break down the process step by step:
Understanding the Image and Subtitles:
    The image shows two individuals at what appears to be a public event or protest. Both individuals are wearing white t-shirts with the text "NO JUSTICE NO PEACE" and images of Trayvon Martin. The English subtitle reads: "Trayvon Martin's Ghost Allegedly Intervenes in Atlanta Protest, Disrupting Police Investigation with Mysterious Messages." The Chinese subtitle reads: "据称，特雷沃恩·马丁的鬼魂介入了亚特兰大的抗议活动，并用神秘信息扰乱了警方的调查。"
Content Consistency Analysis:
    The presence of Trayvon Martin's image on the t-shirts implies a connection to his legacy and possibly a protest related to social justice. The English subtitle suggests an element of paranormal activity with Trayvon Martin's ghost intervening in a protest, which introduces a supernatural claim. The Chinese subtitle translates to: "It is claimed that Trayvon Martin's ghost intervened in the Atlanta protest and disrupted the police investigation with mysterious messages." Both subtitles imply that Trayvon Martin's ghost is an active participant in disrupting police investigations during a protest.
Identifying Misalignment:
    The image alone does not provide any indication of supernatural elements or ghostly interventions. Protest imagery is common and usually focuses on political or social issues rather than paranormal claims. The subtitles contain sensational elements ("ghost," "mysterious messages") that are not supported by standard imagery of protests or justice-related events. The manipulation or misalignment appears to be in the form of exaggerated and unsupported claims about supernatural involvement in a political protest.
Conclusion:
    The image is of a protest with people wearing shirts related to Trayvon Martin, but there is no visual evidence to support the supernatural claims in either subtitle. Both the English and Chinese subtitles introduce elements (the ghost intervention and mysterious messages) that are not depicted or suggested by the image. Hence, the captions are not truthful representations of the image content, leading to the classification of "Both Misaligned."
    No bounding boxes are needed as the manipulation is in the captions, not the image itself.
</think>
<answer>{"classification": "Both Misaligned", "region": {"bbox":[]}}</answer>
```

English Subtitle: Trayvon Martin's Ghost Allegedly Intervenes in Atlanta Protest, Disrupting Police Investigation with Mysterious Messages

Chinese Subtitle: 据称，特雷沃恩·马丁的鬼魂介入了亚特兰大的抗议活动，并用神秘信息扰乱了警方的调查

Figure 17: Case of Both Misaligned.

