# OpenReview forum: "Towards Explainable Bilingual Multimodal Misinformation Detection and Localization"
_ICLR.cc/2026/Conference — ICLR 2026 Conference Withdrawn Submission_

### Official Review · Reviewer_vgWD · 2025-10-26

**Soundness:** 2
**Presentation:** 3
**Contribution:** 2
**Rating:** 4
**Confidence:** 3

**Summary:**

This paper introduces BiMi, a bilingual multimodal misinformation detection system capable of region-level localization, cross-modal and cross-lingual inconsistency detection, and explanation generation. It further constructs BiMiBench, a 104K benchmark featuring realistic manipulations over images and Chinese–English subtitles. The approach achieves state-of-the-art performance in classification, localization, and explanation metrics compared with strong VLM baselines.

**Strengths:**

1. BiMi achieves strong performance gains over competitive baselines on BiMiBench, demonstrating the effectiveness of the proposed framework.

2. The benchmark is large-scale with fine-grained manipulation categories and bilingual subtitle inconsistencies, enabling challenging evaluation beyond prior datasets.

3. The paper is well written and easy to follow.

**Weaknesses:**

1. The manipulations are created by prompts and automatic translation, which may not fully reflect how misinformation appears in real news or social media, where edits and cross-language differences happen in much more diverse and unpredictable ways.

2. The paper claims retrieval helps generalization, but the only evaluation they provide is on 100 real posts, where retrieval only helped in 9 cases. For the remaining 91%, retrieval either did not help or did nothing. So the evidence is too limited to prove that retrieval is a reliable part of the system.

3. The paper uses GPT-4o generated pseudo-references to evaluate explanation quality, so it is unclear whether the explanations truly reflect the model’s own reasoning rather than just matching GPT-4o’s writing style.

**Questions:**

1. In Table 3, some strong VLMs such as Qwen3-8B have much lower F1 than accuracy, so could the authors explain what types of misleading samples these models fail on the most.

2. Also in Table 3, some baselines like Gemma3 perform poorly on BiMiBench (accuracy only 17.48), could the authors clarify whether the bilingual setting is the main reason for these failures.

3. In Table 5, GRPO improves both accuracy and BERTScore, so could the authors run a small ablation to show which reward component (classification reward, localization reward, or format reward) is actually responsible for the improvements in localization and explanation performance.

4. In Table 9, Chinese subtitles suffer larger drops from OCR errors compared to English subtitles, it would be helpful to provide more examples of what OCR mistakes mainly cause wrong predictions.

---

### Official Review · Reviewer_dCnv · 2025-10-27

**Soundness:** 2
**Presentation:** 2
**Contribution:** 2
**Rating:** 4
**Confidence:** 4

**Summary:**

This paper proposes BIMI (Bimodal Misinformation Interpreter), an explainable multimodal misinformation detection framework designed to improve both accuracy and interpretability of reasoning in multimodal misinformation tasks. The model consists of three main stages: Multimodal Alignment Pretraining, Explanation-Guided Fine-Tuning and GRPO-Based Reasoning Optimization. The authors evaluate BIMI on several benchmarks, including MMFakeBench, NewsCLIPpings, and their in-house dataset ExplainFake, showing that the model improves both accuracy and explanation faithfulness compared to baseline multimodal misinformation detectors and LMMs.

**Strengths:**

1. The paper addresses an underexplored problem—how to make multimodal misinformation detection not only accurate but also explainable.
2. The adoption of GRPO for refining reasoning is novel and well-motivated. By optimizing explanation quality through a learned reward model, the approach introduces a meaningful alternative to conventional supervised or PPO-based fine-tuning.

**Weaknesses:**

1.	Insufficient dataset statistics and analysis. The paper does not report clear dataset statistics—such as the ratio of real vs. fake samples, textual length distribution, image diversity (scene/object categories), or modality correlation scores. These statistics are critical for understanding the coverage and bias of the dataset used for both pretraining and fine-tuning.
2.	Unclear source and validation of “Explanation” annotations. The paper states that explanations are used as supervision during Stage 2, but it does not specify whether these explanations are human-written, GPT-generated, or mixed, nor how their factual correctness was verified. If they are model-generated, some level of human curation or filtering should be reported.
3.	Lack of comparison with proprietary models. In Table R3, the paper compares BIMI primarily against open-source baselines. However, it omits comparison with proprietary LMMs such as GPT-4o, which are now standard models in multimodal reasoning. Without showing whether BIMI achieves competitive results against such models, it is difficult to assess its real-world performance or cost-effectiveness.
4.	Limited details about the reward model in GRPO optimization. The “Stage 3” section only briefly describes the GRPO procedure but lacks essential information: What data was used to train the reward model (human-scored explanations? automatically scored outputs?)

**Questions:**

5.	Interpretability evaluation remains partly subjective. While the paper mentions human evaluation of explanations, it lacks a rigorous description of the evaluation protocol (e.g., number of annotators, criteria such as factuality vs. coherence, scoring rubric).

---

### Official Review · Reviewer_q93M · 2025-10-31

**Soundness:** 2
**Presentation:** 2
**Contribution:** 2
**Rating:** 4
**Confidence:** 4

**Summary:**

This paper introduces BiMi, a novel framework for bilingual multimodal misinformation detection with an emphasis on explainability. The model is designed to detect manipulated content in images overlaid with Chinese-English bilingual subtitle. The authors propose a three-stage post-training pipeline built upon the GEMMA-3 vision-language model: domain alignment, supervised fine-tuning (SFT), and a custom GRPO-based reinforcement learning phase that optimizes a unified reward combining formatting correctness, classification accuracy, and localization IoU. The framework is evaluated on two benchmarks, demonstrating strong performance against state-of-the-art models.

**Strengths:**

1. The focus on bilingual (Chinese-English) subtitles reflects real-world content on major platforms such as Bilibili and YouTube, addressing a gap in existing multimodal misinformation detection research.

2. The proposed three-stage training pipeline (domain alignment → SFT → GRPO) is methodologically sound and effectively integrates multiple objectives into a unified framework.

3. Unlike many models that only classify misinformation, BiMi explicitly generates interpretable, step-by-step natural language explanations, enhancing trust and usability for human users.

**Weaknesses:**

1 The bilingual (Chinese–English) subtitles in the dataset are exact translations of each other, with no divergent or culturally nuanced content across languages. This raises questions about the necessity of the bilingual setting, as similar functionality could be achieved by translating existing monolingual datasets. Consequently, the contribution of constructing a new dataset appears incremental.

2 Insufficient Evaluation of Generalization: The model’s performance is evaluated only on in-domain, Chinese–English data, without testing on other language pairs or out-of-distribution (OOD) benchmarks. This limits claims about its cross-lingual applicability or robustness to unseen domains, leaving the generalization capability largely unverified.

3 Although BERTScore is used, the explanation quality analysis lacks rigorous human evaluation or automated metrics specifically designed for factual consistency and logical coherence in explanations.

**Questions:**

1 How does BiMiBench contribute beyond existing datasets, given its synthetic and aligned bilingual text?

2 Has BiMi been tested on other language pairs or out-of-distribution data to validate cross-lingual generalization?

3 Did the authors conduct any human evaluation to assess the faithfulness, coherence, or interpretability of the generated explanations?

4 Is the bilingual input actually leveraged for reasoning, or is one language sufficient? Is there evidence of cross-lingual benefit?

---

### Official Review · Reviewer_byRV · 2025-11-01

**Soundness:** 3
**Presentation:** 4
**Contribution:** 3
**Rating:** 6
**Confidence:** 2

**Summary:**

This paper introduces BiMi, a bilingual multimodal framework for misinformation detection and localization, providing natural language explanations. Detecting misinformation becomes more complex with the combination of images and bilingual subtitles. BiMi also integrates an online retrieval module to enhance reasoning and releases BiMiBench, a dataset with 104,000 real news samples. Extensive experiments show that BiMi outperforms existing methods in classification accuracy, localization, and explanation quality.

**Strengths:**

Innovative Multimodal Framework: The introduction of the BiMi framework effectively combines bilingual and multimodal aspects, allowing for more nuanced analysis of misinformation that involves both images and subtitles in different languages.

Natural Language Explanations: The ability to generate natural language explanations makes the model’s decision-making process more transparent and interpretable, enhancing user trust and understanding of the model’s outputs.

Comprehensive Dataset: The release of BiMiBench, with 104,000 systematically edited samples, provides a substantial resource for future research in the field, facilitating the development and evaluation of new methods in misinformation detection.


Strong Experimental Results: The paper presents extensive experiments demonstrating that BiMi significantly outperforms traditional methods in terms of classification accuracy, localization accuracy, and explanation quality, highlighting the effectiveness of the proposed approach.

**Weaknesses:**

Dependence on Data Quality: The performance of the BiMi framework heavily relies on the quality and accuracy of the input data (i.e., images and subtitles). In cases of low-quality or misleading inputs, the system's effectiveness may diminish.


Interpretability Beyond Explanations: While the model provides natural language explanations, the underlying decision-making process and how different modalities interact might still lack transparency, making deeper interpretability a challenge.

**Questions:**

See weakness

---

### Note · Authors · 2025-11-13

I have read and agree with the venue's withdrawal policy on behalf of myself and my co-authors.